# Inflammatory Myofibroblastic Tumour of the Urinary Bladder in a Middle-Aged Man—A Case Report of an Unusual Localization of a Rare Tumour

**DOI:** 10.3390/medicina59040791

**Published:** 2023-04-19

**Authors:** Nebojsa Prijovic, Veljko Santric, Uros Babic, Branko Stankovic, Miodrag Acimovic, Milica Cekerevac, Gorana Nikolic, Bojan Cegar

**Affiliations:** 1Clinic of Urology, University Clinical Center of Serbia, 11000 Belgrade, Serbia; nebojsa.prijovic@yahoo.com (N.P.); veljkosantric@yahoo.com (V.S.); urosb2001@yahoo.com (U.B.); bstank@gmail.com (B.S.); miodrag.acimovic@med.bg.ac.rs (M.A.); 2Faculty of Medicine, University of Belgrade, 11000 Belgrade, Serbia; gorana.nikolic03@gmail.com; 3Department of Pathology, University Clinical Centre of Serbia, 11000 Belgrade, Serbia; milicach@yahoo.com; 4Institute of Pathology, Faculty of Medicine, University of Belgrade, 11000 Belgrade, Serbia

**Keywords:** inflammatory myofibroblastic tumour, urinary bladder, partial cystectomy

## Abstract

Inflammatory myofibroblastic tumour (IMT) is a rare tumour with an intermediate biological behaviour. It usually occurs in children and adolescents, primarily in the abdomen or lungs. Histopathologically, IMT consists of spindle cells, i.e., myofibroblasts, and a variable inflammatory component. Localization in the urinary bladder is rare. We are presenting a rare case of IMT in the bladder in a middle-aged man treated by partial cystectomy. A 62-year-old man consulted a urologist because of haematuria and dysuric disturbances. A tumorous mass was detected by an ultrasound in the urinary bladder. CT urography described the tumorous mass at the dome of the urinary bladder measuring 2 × 5 cm. A smooth tumorous mass was cystoscopically observed at the dome of the urinary bladder. Transurethral resection of the bladder tumour was performed. Histopathological analysis of the specimen identified spindle cells with a mixed inflammatory infiltrate; immunohistochemical findings showed positivity for anaplastic lymphoma kinase (ALK), smooth muscle actin (SMA) and vimentin. A histopathological diagnosis of IMT was established. It was decided that the patient would undergo a partial cystectomy. A complete excision of the tumour from the dome of the urinary bladder with surrounding healthy tissue was performed. Histopathological and immunohistochemical findings of the sample confirmed the diagnosis of IMT, without the presence of the tumour at the surgical margins. The postoperative course went smoothly. IMT is a rare tumour in adults, especially localised in the urinary bladder. IMT of the urinary bladder is difficult to distinguish from urinary bladder malignancy both clinically and radiologically, as well as histopathologically. If the location and size of the tumour allow it, bladder-preserving surgeries such as partial cystectomy represent a reasonable modality of operative treatment.

## 1. Introduction

Inflammatory myofibroblastic tumour (IMT) is a very rare tumour from the group of fibroinflammatory disorders [1]. IMT is a tumour with an intermediate biological potential and a low possibility of recurrence and metastasis [2]. The etiology of IMT is not completely clear. It most often occurs in childhood or adolescence, although it can also occur in old age [3]. Predominant localizations of IMT are lungs and abdominal organs, but it can occur in any localization [4]. Data from the literature indicate that IMT (previously described as an inflammatory pseudo-tumour) occurs in the lungs of one third of children patients, while two thirds have extrapulmonary localization of the disease [5]. Histopathologically, IMT consists of spindle cells, i.e., myofibroblasts, and a variable inflammatory component [6]. Radiologically, these tumours cannot be distinguished from malignant tumours based on the findings of imaging tests, especially since they appear as large tumours [1,7].

We present an unusual occurrence of IMT in a middle-aged man in the urinary bladder as a rare localization for this tumour. The patient’s histopathological diagnosis was established after transurethral resection of the tumour and after the tumorous mass was completely removed by performing a partial cystectomy.

## 2. Detailed Case Description

A 62-year-old male patient, a non-smoker, consulted a urologist because of dysuric disorders and haematuria. The patient denied fever, weakness and weight loss. The patient denies previous urinary infections, injuries and surgical interventions of the urinary tract. An echosonographic examination was performed in which a tumorous mass in the urinary bladder was observed; therefore, a computerized tomography (CT) examination was indicated. On the CT urography, a tumorous mass was observed at the dome of the urinary bladder, measuring 2 × 5 cm. Perivesical tissue of increased attenuation reached the parietal peritoneum. Lymph nodes in the pelvis and abdomen were with normal findings on the CT scan. The patient was referred to our institution, the Urology Clinic of the University Clinical Center of Serbia. Upon admission to our institution, laboratory analyses indicated the following: white blood cells 5.5 × 10^9^/L, red blood cells 4.3 × 10^12^/L, haemoglobin 136 g/L and platelets 205 × 10^9^/L. A cystoscopic examination was performed, in which a tumorous mass was observed at the junction of the front wall and the dome of the urinary bladder, with a diameter of about 4 cm, a wide base and a smooth surface. According to the above-mentioned characteristics, we first thought that it was a tumour of urachus origin. We decided to initially perform a transurethral resection of the bladder tumour (TURBT) in order to obtain a histopathological finding that would direct us to further treatment of the patient. TURBT was performed when the indicated lesion was resected to the level of the wall. The specimen was sent for histopathological analysis to an experienced uropathologist. The obtained histopathological findings showed that the tumorous mass consists of spindle cells of discohesive growth, myxomatous and inflammatory areas of mixed inflammatory infiltrate. No mitoses or necrosis were observed in the tumour. Immunohistochemical findings showed positivity for anaplastic lymphoma kinase (ALK), smooth muscle actin (SMA) and vimentin. Figure 1 shows the histopathological and immunohistochemical features of IMT in our patient’s case The pathologist’s conclusion was that it was an inflammatory myofibroblastic tumour.

Considering the findings of the CT urography and the histopathological diagnosis, it was decided that the patient should undergo a partial cystectomy. Two months after a transurethral resection of the bladder tumour, the patient was hospitalised again in our institution for planned further surgical treatment. Upon admission, repeated CT urography showed inhomogeneous thickening of the anterior wall and the dome on the left, about 22.5 mm thick with dystrophic calcifications present, unchanged perivesical fat tissue and no pathological lymphadenopathy in the abdomen and pelvis. (Figure 2) Under general anaesthesia, after laparotomy and access to the urinary bladder, a 3 cm-sized tumour was identified at the dome of the urinary bladder, which was resected in its entirety with the surrounding healthy tissue.

The intraoperative findings are shown in Figure 3. During the partial cystectomy, the tumour was completely resected with the surrounding macroscopically unchanged tissue about 10 mm wide. We defined the resected surrounding unchanged tissue around the tumour as the surgical margins. The bladder was sutured in two layers with placement of a three-way urinary catheter into the urinary bladder and a drain into the pouch of Douglas.

The tumour specimen (Figure 4) and the surgical margin specimen were sent for histopathological analysis.

The postoperative course went smoothly, without complications. The patient had his drain displaced and then also his urinary catheter on the 6th postoperative day. The patient was discharged home on the 6th postoperative day, afebrile, in good general condition and with regular urination. The wound healed per primam. Histopathological analysis of the specimen after partial cystectomy confirmed the diagnosis of IMT—the presence of spindle cells with a storiform pattern without nuclear atypia with the presence of a mixed inflammatory infiltrate. The presence of tumours was not observed in the surgical margins. Immunohistochemical analysis showed positivity for ALK, SMA and vimentin, while negativity was observed for epithelial membrane antigen (EMA). The control ultrasound and cystoscopic examination were performed 3 months after the surgery; there were no signs of tumour recurrence. The patient subjectively felt well with no symptoms of the lower urinary tract and haematuria.

## 3. Discussion

Inflammatory myofibroblastic tumour is a very rare tumour with intermediate biological behaviour and low risk of recurrence and metastasis [2]. Recurrence of IMT can occur in up to 25% of cases, while the presence of distant metastases is described in less than 2% of cases. [8] IMT usually occurs in children and young adults, although it has been reported to occur in all age groups [3]. It is estimated that the prevalence of this tumour in the world is between 0.04% and 0.7% regardless of gender and race [9,10]. The real prevalence of IMT in the population cannot be accurately estimated considering the changes in the nomenclature and classification of fibroinflammatory disorders [9]. IMT has been described so far under various fibroinflamatory disorders and soft tissue sarcomas [11]. Although the pulmonary localization of IMT was first described, which occurs in about a third of cases [5,12], the most common localization is the abdominal cavity, especially the mesentery, omentum or retroperitoneum [4]. Given its less frequent occurrence in middle-aged or elderly patients, there can be difficulties in diagnosing this tumour [8]. Given the more frequent occurrence of this tumour in the visceral organs and deep soft tissues of the abdomen, pelvis and retroperitoneum, the diagnosis of IMT in other locations such as the urinary bladder should be made by excluding other histopathologically related disorders [8]. Given that urothelial carcinoma is the most common type of urinary bladder tumour in the elderly population, diagnosing other histologies in adults is a challenge for the pathologist who is faced with the question of whether it is one of the many faces of urothelial carcinoma or another pathology [13].

The etiology of IMT is not completely clear. It is believed that previous urinary infections, trauma, surgery or immunosuppression can play a role in the development of IMT of the urinary bladder [14,15], although there are also cases where etiological factor for the development of IMT cannot be identified [16]. In our case, the patient denies previous urinary infections, urinary tract instrumentation and surgery. He also denies the existence of previous urinary tract trauma.

IMT is manifested by variable and non-specific symptomatology depending on its localization. The rare cases of IMT in the urinary bladder described so far have shown that IMT can be manifested by haematuria and/or dysuric disorders [16,17], i.e., complaints that our patient also had. Fever, malaise and weight loss are often present, occurring in 15–30% of patients [18,19]. In our case, the patient denied the mentioned constitutional symptoms. Microcytic anaemia and thrombocytosis can be present in laboratory analyses in patients with IMT [18,19], which was not the case in our patient, in whom all laboratory parameters were within normal limits.

Radiologically, IMT is difficult to distinguish from urinary bladder malignancy. IMT an unusual radiologically resemble both benign and malignant tumours [20]. In our patient, the initial CT examination showed the existence of a tumorous mass at the dome of the urinary bladder with a diameter of about 5 cm associated with clinically present haematuria. Given that these tumours radiologically appear as large tumours of the urinary bladder, and without a previous history of TURBT, such a finding can seem worrisome [7]. In our case, the initial thought of the competent urologist was that it was a tumour of urachus origin. Considering the specific way of operative treatment of urachus tumour, it was decided to perform TURBT first to obtain a histopathological diagnosis.

An experienced uropathologist diagnosed IMT of the urinary bladder in our patient. IMT is histopathologically characterised by the proliferation of spindle cells in a myxoid and colloid stroma with an inflammatory infiltrate that may consist of plasma cells and lymphocytes, with rarer eosinophils and neutrophils [8]. Mitotic activity in IMT is low, and atypical mitoses are rare [21]. There are no specific immunohistochemical markers for IMT. The findings so far show strong positivity for vimentin, as well as for SMA and ALK in about 50% of cases [22]. ALK is a tyrosine kinase receptor first described in anaplastic large cell lymphoma [23]. The discovery of the ALK gene rearrangement was a milestone in defining IMT as a neoplasm [24]. The resulting overexpression of the ALK protein is detectable by conventional immunohistochemical methods [22,25,26]. The immunohistochemical finding of ALK is relatively specific for IMT in the spectrum of histopathologically similar fibroblastic–myofibroblastic tumours [22,26]. Despite the importance of ALK positivity in establishing the diagnosis of IMT, no difference was observed in the clinical behavior of IMT in case of ALK positivity and negativity [27]. In addition, ALK rearrangement was observed to be more common in younger patients with IMT compared to older ones [28].

Considering the received histopathological findings as well as the repeated findings of CT urography, it was decided that the patient should undergo a partial cystectomy. According to the guidelines of the European Society for Medical Oncology (ESMO), the standard in the treatment of localised tumours is surgical treatment [29]. En bloc tumour resection with negative surgical margins is recommended as a standard surgical procedure, which implies removal of the tumour in one specimen with a rim of normal tissue around it. Additionally, it is recommended that the procedure be performed by a well-trained surgeon. Given that current findings indicate that IMT of the urinary bladder has a benign course, conserving treatment options such as TURBT and partial cystectomy are reasonable treatment options [27]. In rare cases, IMT can show local aggressiveness with involvement of the prostate and other pelvic structures [30], and in rare cases radical cystectomy is necessary [27]. Given the risk of recurrence of IMT [17], in our case we performed a partial cystectomy, by which we resected the tumorous mass as a whole with the surrounding healthy tissue. After complete tumour resection, no adjuvant therapy is required [31]. Given that the histopathological analysis of the specimen of our patient after partial cystectomy showed that the tumour was not present at the surgical margins, we decided for surveillance of the patient. Follow-up of patients after surgical removal of IMT is not precisely defined. Given the complete resection of the tumour without the presence of tumour on the surgical margins, the further plan of monitoring the patient is regular ultrasound and cystoscopic controls. In case of abnormal findings of these diagnostic procedures, we would additionally perform CT urography.

## 4. Conclusions

An inflammatory myofibroblastic tumour is a rare tumour in adults, especially in the urogenital tract. IMT of the urinary bladder is difficult to distinguish from tumours with a higher malignant potential both clinically and radiologically, as well as histopathologically. If the location and size of the tumour allow it, bladder-preserving methods such as partial cystectomy represent a reasonable modality of operative treatment.

## Figures and Tables

**Figure 1 medicina-59-00791-f001:**
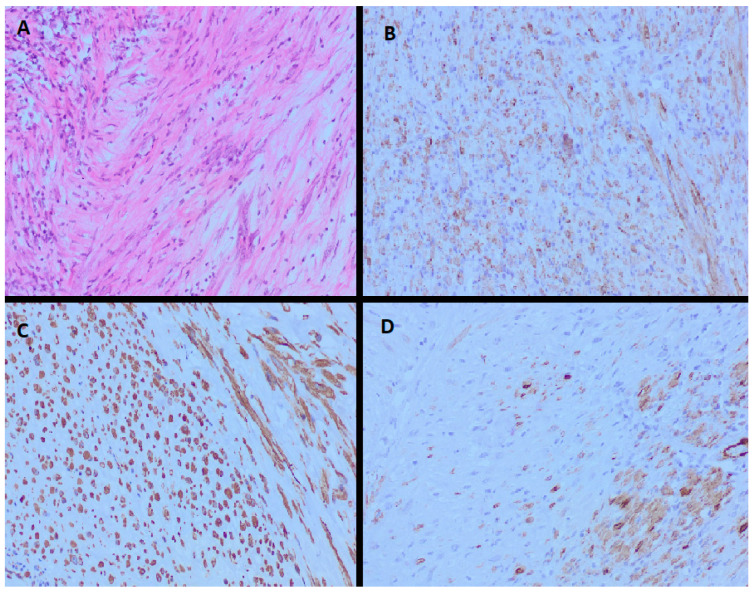
Histopathological findings showed that the tumorous mass consists of spindle cells with mixed inflammatory infiltrate (**A**). Immunohistochemical findings showed positivity for anaplastic lymphoma kinase (ALK) (**B**), vimentin (**C**) and smooth muscle actin (SMA) (**D**).

**Figure 2 medicina-59-00791-f002:**
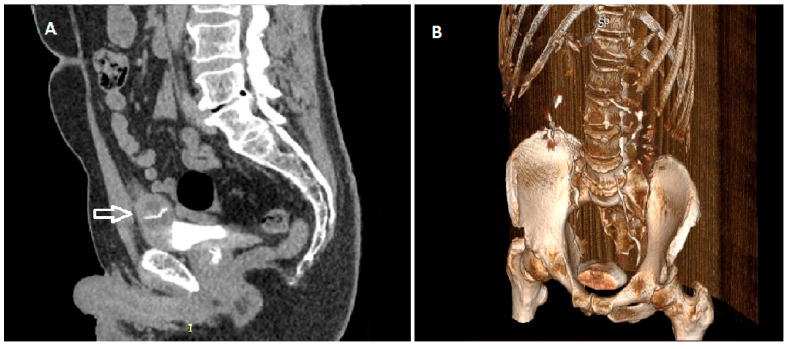
CT urography shows a tumour at the dome of the urinary bladder (arrow)—sagittal plane (**A**) The 3D reconstruction also shows a lesion at the dome of the urinary bladder (**B**).

**Figure 3 medicina-59-00791-f003:**
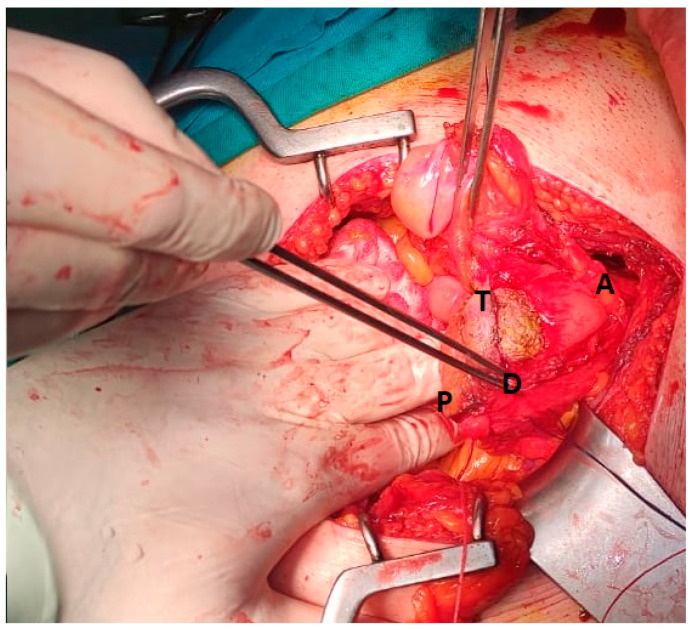
Intraoperative finding and the performance of the tumour resection from the dome of the urinary bladder. A—anterior wall of the urinary bladder; D —the dome of the urinary bladder; P—posterior wall of the urinary bladder; T—tumour.

**Figure 4 medicina-59-00791-f004:**
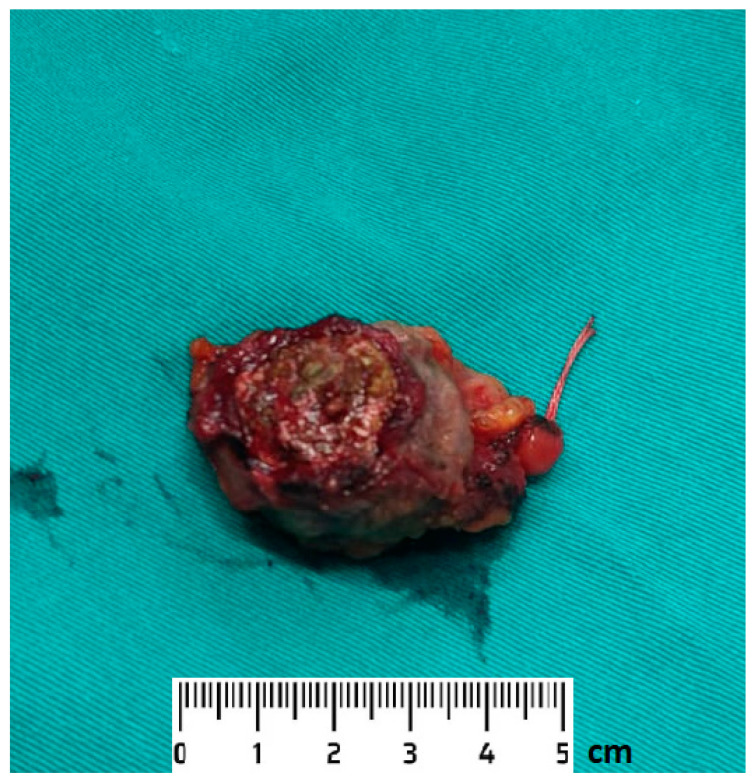
Excised bladder tumour specimen.

## Data Availability

All data shown in this study are included in this published article.

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
