# Peer review of "Inflammatory Myofibroblastic Tumour of the Urinary Bladder in a Middle-Aged Man—A Case Report of an Unusual Localization of a Rare Tumour"

_medicina, 2023, doi:10.3390/medicina59040791_

Round 1

Reviewer 1 Report

. There are no specific immunohistochemical markers for IMT. The 141 findings so far show strong positivity for vimentin, as well as for SMA and ALK in about 142 50% of cases

Vimentin has limited/no value in this context, would not mention it

The statement 'There are no specific immunohistochemical markers for IMT' is confusing, because with the appropriate morphology, expression of ALK is essentially diagnostic, this should be clarified

Author Response

Dear reviewer,

Thank you for your comment. In the revised manuscript, we described in more detail the importance of ALK in the diagnosis of IMT.

Sincerelly yours,

Bojan Cegar

Reviewer 2 Report

Authors reported the rare case of IMT in urinary bladder. Although the report has some interesting findings, current version of the manuscript does not reach the standards of publication in the medicina.

Indeed, IMT in the urinary bladder is rare. However, there have already been many reports of IMT in urinary bladder previously. Thus, it is not so novel just to report case presentation.

#1 A and B in figure 2 are essentially the same image. Do you have images of other modalities? Also, a cystoscopy image would be better.

#2 Indicate the anatomical structures in Figure 3 with arrows.

#3 Add a scale to Figure 4.

#4 Did this patient have a history of trauma or infection? Discuss the probable cause for this patient to develop IMT in urinary bladder.

#5 How was “surrounding healthy tissue” determined as the resection margin around the tumor?

#6 Please mention this patient's prognosis and future follow-up methods.

#7 Many of the references are over ten years old. It is better to cite more recent literature.

Author Response

Dear reviewer,

We appreciate your comments and suggestions. In the revised manuscript, we have tried to make changes in accordance with your comments.

A and B in figure 2 are essentially the same image. Do you have images of other modalities? Also, a cystoscopy image would be better.

We agree that Figure 1 shows the same tumour, except that the tumor is shown in the computed tomography in the sagittal plane and in the 3D reconstruction. Unfortunately, we do not have images of other diagnostic modalities.

 Indicate the anatomical structures in Figure 3 with arrows.

Thank you for this suggestion. We marked the structures in Figure 3.

Add a scale to Figure 4.

Thank you for this comment. We have made the changes listed in Figure 4.

Did this patient have a history of trauma or infection? Discuss the probable cause for this patient to develop IMT in urinary bladder.

Thank you and we appreciate this comment. The patient had no history of infection or injury. In the revised manuscript, we have provided this information in the case presentation and discussion.

How was “surrounding healthy tissue” determined as the resection margin around the tumor?

Thank you for this comment. During the partial cystectomy, the tumor was completely resected with the surrounding macroscopically unchanged tissue about 10 mm wide. We defined the surrounding unchanged tissue around the tumor that we resected as surgical margins.

Please mention this patient's prognosis and future follow-up methods.

Thank you for this comment. We have included corrections in the text related to the planned follow-up of the patient.

Many of the references are over ten years old. It is better to cite more recent literature.

Thanks for this comment. We tried to use relatively recent sources, but we had to cite earlier works related to the first described cases of IMT and its earlier names in classifications. Almost half of the references are under 10 years old.

In the hope that you will respond positively to our corrections, I thank you in advance.

Sincerelly yours,

Bojan Cegar

Round 2

Reviewer 2 Report

well corrected